# Effect of Zinc and Severe Plastic Deformation on Mechanical Properties of AZ61 Magnesium Alloy

**DOI:** 10.3390/ma17071678

**Published:** 2024-04-05

**Authors:** Song-Jeng Huang, Sheng-Yu Wu, Murugan Subramani

**Affiliations:** Department of Mechanical Engineering, National Taiwan University of Science and Technology, No. 43, Section 4, Keelung Rd, Da’an District, Taipei 10607, Taiwan; shengyou1999@gmail.com (S.-Y.W.); smurugan2594@gmail.com (M.S.)

**Keywords:** AZ61 magnesium alloy, ECAP, heat treatment, mechanical properties, corrosion resistance

## Abstract

This study investigates the effects of zinc (4 wt.%) and severe plastic deformation on the mechanical properties of AZ61 magnesium alloy through the stir-casting process. Severe plastic deformation (Equal Channel Angular Pressing (ECAP)) has been performed followed by T4 heat treatment. The microstructural examinations revealed that the addition of 4 wt.% Zn enhances the uniform distribution of β-phase, contributing to a more uniformly corroded surface in corrosive environments. Additionally, dynamic recrystallization (DRX) significantly reduces the grain size of as-cast alloys after undergoing ECAP. The attained mechanical properties demonstrate that after a single ECAP pass, AZ61 + 4 wt.% Zn alloy exhibits the highest yield strength (YS), ultimate compression strength (UCS), and hardness. This research highlights the promising potential of AZ61 + 4 wt.% Zn alloy for enhanced mechanical and corrosion-resistant properties, offering valuable insights for applications in diverse engineering fields.

## 1. Introduction

In the aerospace, automotive, and green energy industries, the concept of employing large structural components has gained widespread acceptance due to their immense potential in enhancing operational efficiency, reducing carbon dioxide emissions, and achieving light-weighting. The pursuit of these characteristics has driven extensive research and development efforts [1,2,3,4]. Magnesium alloys, being the lightest structural metals, are considered a powerful choice for achieving industrial light-weighting. With a density of only two-thirds that of aluminum alloys and one-fourth that of steel, magnesium alloys offer significant advantages in light-weighting.

Furthermore, magnesium alloys possess advantageous features such as high damping for shock absorption, efficient heat dissipation, and recyclability. For example, in the automotive industry, magnesium alloys find application in various components like steering wheels, seat frames, dashboard frames, and gearbox casings [5,6].

Despite these merits, challenges persist in the application of magnesium alloys. Issues such as corrosion resistance and strength need further improvement to meet the application requirements under high-temperature conditions in the automotive engine compartment [7,8,9]. Additionally, the processing performance of magnesium alloys requires enhancement to replace traditional steel and aluminum materials in a broader range of automotive components [10]. The automotive industry, grappling with the need for increased safety and luxury features, faces the challenge of minimizing overall vehicle weight [11]. Therefore, the development of new magnesium alloys and improved magnesium-alloy-processing techniques is crucial for achieving automotive light-weighting [12,13].

To address these challenges, various strategies have been devised, including alloying, heat treatment, and the application of severe plastic deformation (SPD) [14,15,16]. Among SPD techniques, Equal Channel Angular Pressing (ECAP) has been demonstrated to yield the most significant grain-refinement effects in magnesium alloys [15,17]. The ECAP process generates a substantial amount of cumulative strain, playing a crucial role in the microstructure and texture evolution of magnesium alloys. In the ECAP process, materials are typically preheated to approximately half of their melting point [18], as performed in this study at 350 °C. Given that magnesium’s hexagonal close-packed (HCP) crystal structure has only two independent slip systems at low temperatures, dynamic recrystallization (DRX) becomes crucial for the microstructure and texture evolution of magnesium alloys subjected to ECAP treatment [19,20].

In the initial grains and twin boundaries, new very small grains form in a specific crystallographic orientation due to DRX [21]. Aqeel Abbas et al. (2020) employed AZ91 magnesium alloy as the base material [22], added 1 wt.% tungsten disulfide (WS2) as a reinforcement phase, and used mechanical stirring casting to prepare the material. The material was homogenized at 410 °C for 24 h and annealed at 200 °C. The severe plastic deformation process chosen was ECAP, and the results showed that after 4 ECAP passes, the grain size had reduced to 0.2 mm, with hardness and tensile yield strength increased by 20.45% and 103.5%, respectively. Moreover, 4 ECAP passes exhibited the highest ultimate tensile strength of 324.8 MPa.

This study investigates the impact of alloying (increasing Zn content), solution heat treatment (T4 heat treatment), and severe plastic deformation (ECAP) on the strength enhancement of AZ61 magnesium alloy. There is a lack of comprehensive research on the combined effects of heat treatment and ECAP on the mechanical properties of AZ61 Mg alloy. Thus, this investigation delves into the microstructure analysis, mechanical properties, and corrosion resistance of both AZ61 alloy and AZ61 + 4 wt.% Zn, considering the influence of heat treatment and ECAP.

## 2. Materials and Methods

### 2.1. Materials Preparation

A commercial AZ61 magnesium alloy (mainly composed of Al-6% and Zn-1%) (Bo Tao Nanotechnology Co., Ltd., Taipei, Taiwan) served as the matrix component. The chemical composition of AZ61 magnesium alloy is shown in Table 1. Zn powder (4 wt.%) (Emperor Chemical Co., Ltd., Taipei, Taiwan) particles were used as reinforcements.

The alloys were produced using a stir-casting technique as illustrated in Figure 1. Initially, the base material and reinforcement were placed into the crucible furnace. When the temperature reached 400 °C, a mixture of SF6 + CO2 gas was added to prevent burning. Subsequently, the temperature was raised to 600 °C using Ar gas to prevent oxidation. Once the temperature reached 760 °C, it was held for ten minutes. Following this, the mixture was stirred rapidly for 20 min at 250 rpm to disperse the reinforcement particles. After completing the stirring process, the plunger was raised to allow the molten metal to flow into the mold. The material was then extracted and left to solidify through natural air cooling. Finally, the last casting was removed, and the mains power supply was shut down [23]. Two different types of cylindrical ingots were prepared using the same method and fabrication process: 1) AZ61 and 2) AZ61 + 4% Zn.

Consequently, the ingots were cut into bars measuring 11.5 mm × 11.5 mm × 75 mm, to facilitate subsequent processing steps, including specific heat treatment and Equal Channel Angular Pressing (ECAP) shown in Figure 2.

In this study, we employed the T4 solid solution treatment to fabricate our specimens. Initially, the samples underwent a 24 h heat treatment at 380 °C to achieve a specific solid solution effect. Subsequently, the treated specimens were directly quenched at room temperature to ensure the integrity of the heat-treatment process. The purpose of this heat-treatment procedure is to tailor the material’s crystal structure and mechanical properties to meet the requirements of our research. Following the heat treatment, the ECAP fabrication process (Figure 2), was carried out on a preheated mold of the ECAP device at 350 °C for 40 min. Subsequently, a single pass of ECAP was conducted with a channel angle of 120 degrees and a pressing speed of 20 mm/min.

### 2.2. Microstructural Characterization

X-ray Diffraction (XRD) patterns were analyzed using the Bruker D2 PHASER X-ray (Bruker Co., Boston, MA USA) Diffractometer. Operating at 45 kV, the anode copper (monochromatic Cu Kα) source provided detailed crystalline-structure insights in the 20- to 80-degree range at a 0.04 °/s scan rate. Subsequent data analysis was performed utilizing the Bruker EVA 5.2 software (Bruker Co., Boston, MA, USA). The obtained diffraction patterns were compared with the data from the Joint Committee of Powder Diffraction Standards (JCPDS) database, a widely recognized reference, to confirm the metallic phase composition and identify variations in diffraction planes.

Optical microscopy (OM) analysis was performed using the Zeiss Axiotech 25HD (Oberkochen, Germany) microscope. Prior to observation, the material underwent meticulous sample preparation, including wet grinding with 4000-grit sandpaper and polishing with a mixture of 50 nm aluminum oxide powder and deionized water (1:5 ratio). The optimized surface preparation aimed to achieve a uniform and smooth sample for detailed optical examination. Subsequently, a suitable etchant was applied to unveil the metallographic features, as specified in Table 2 (Emperor Chemical Co., Ltd., Taipei, Taiwan). The microscope facilitated the detailed observation of macroscopic morphology, grain size, and defects.

SEM analysis using the JEOL 7900F FE-SEM (Japan) allowed high-magnification observation of the material’s surface morphology and microstructural features. Equipped with Energy Dispersive Spectrometry (EDS), it allowed comprehensive elemental analysis, enhancing understanding of the material’s composition and distribution at the microscopic level.

### 2.3. Mechanical Characterization

A Vickers hardness tester FR-1AN (Tokyo, Japan) with VHPro Express software, adhering to ASTM E18–94 standards, was employed to analyze the hardness of AZ61 alloys. Using a 100 gf load, the machine created a rhombic impression on the material’s surface with a pyramidal diamond indenter (136° included angle) and an indentation duration of 10 s. In this experiment, a total of 6 sets of specimens were tested, with 6 different points selected on each specimen for hardness testing, and the average values were calculated.

Compression testing adhered to the ASTM E9 standard, utilizing specimens sized at 20 × 10 × 10 mm^3^. The MTS-810 (MTS systems Corp., Cary, NC, USA) Compression Testing Machine was employed for the experiments. Each specimen underwent three compression tests, and the average values were calculated to determine the material’s properties, including EL (%), YS (MPa), and UCS (MPa).

### 2.4. Corrosion Properties

The experiment adhered to ASTM B117 standards. A 72 h salt spray test was conducted using deionized water containing 5% NaCl to induce corrosion behavior. The testing environment was maintained at a strict temperature of 35 ± 0.5 °C. Through meticulous observation and detailed corrosion-area measurements, this study aimed to evaluate the material’s corrosion resistance and performance under harsh environmental conditions. The specimen dimensions were 20 mm × 10 mm × 10 mm, with one specimen taken for each material. The top surface of the specimens was selected for observation, as it is least affected by contact with the testing platform, thus providing a more accurate assessment of corrosion behavior.

## 3. Results

### 3.1. Phase Characterization

Figure 3 illustrates the X-ray diffraction (XRD) analysis of AZ61 alloy (Figure 3a) and AZ61 + 4 wt.% Zn alloy (Figure 3b). XRD peaks corresponding to the α- Mg_0.97_Zn_0.03_ phase and β-Mg_17_Al_12_ phase are observed in both as-cast AZ61 and AZ61 + 4 wt.% Zn. No zinc containing phase is detected, as zinc has a certain solid solubility in magnesium and thus exists in the form of Mg_0.97_Zn_0.03_ [24]. In the XRD patterns within the 2θ range of 30°–45°, a noticeable decrease in the intensity of β-Mg_17_Al_12_ phase XRD peaks (411), (332) is observed after T4 heat treatment for both AZ61 and AZ61 + 4 wt.% Zn alloys. Moreover, after undergoing ECAP with one pass, only the α—Mg_0.97_Zn_0.03_ phase was observed, indicating homogenization and dissolution of the β-Mg_17_Al_12_ phase during the ECAP process. It is worth noting that this dissolution phenomenon may be attributed to the ECAP process and to heat treatment prior to the ECAP process. These XRD results align with the observations from optical microscopy (OM), as depicted in Figure 4.

In Figure 3, the analysis of the XRD reveals the texture change of the AZ61 alloy after passing through the ECAP channel, which is shown compared to the zero pass. Typically, the basal plane (002) and (100) line of the material are parallel to the extrusion direction [25]. A similar trend is also observed in the SPD-processed samples of AZ61 alloy, where the largest grain distribution aligns parallel to the extrusion direction. From previous studies, it has been noted that after the ECAP process, the peak intensity of the (100) line in magnesium alloy samples increases, resulting in an increase in the alloy’s strength [26]. In Figure 3b, the (100) line becomes more intense than the (002) line in the AZ61 + 4 wt.% Zn alloy after ECAP, exhibiting higher mechanical properties.

### 3.2. Microstructure Evaluation

The OM analysis of the as-cast, homogenized, and ECAP-1 Pass AZ61 alloys are shown in Figure 4a,d depicting the as-cast AZ61 and AZ61 + 4 wt.% Zn, respectively. With an increase in zinc content of 4 wt.% (indicated by yellow arrows), Mg-Zn phases and second phases are observed growing along grain boundaries, with dispersion becoming more uniform. Figure 4b,e represents the AZ61 and AZ61 + 4 wt.% Zn after the homogenization treatment. It reveals that most of the second phase and Zn has dissolved into the α-Mg matrix, but some remnants are still present along the grain boundaries (highlighted by yellow arrows). In particular, the solid solution effect is most pronounced in the AZ61 + 4 wt.% Zn alloy, as evidenced in the results in Figure 4e, indicating that the T4 heat treatment at 380 °C has a higher solid solution effect on Zn.

Figure 4c,f shows the AZ61 and AZ61 + 4 wt.% Zn after one pass of ECAP. Compared to T4 homogenization conditions, the grain size of the samples subjected to ECAP is significantly refined, which can be mainly attributed to the significant contribution of the dynamic recrystallization (DRX) mechanism. The sample with 4 wt.% zinc content shows a more pronounced grain refinement as shown in Figure 4f (highlighted by yellow circles), with larger areas of extensive grain refinement and the generation of more fragmented small grains.

The grain size distribution, as shown in Figure 5, indicates that severe plastic deformation processes improve the microstructure of the alloy, resulting in a significant grain-refinement effect. Figure 5a,d depicts the grain size distribution in the alloy OM images after T4 heat treatment, showing the presence of coarse grains with average sizes of 37.62 ± 8.81 µm (AZ61) and 33.82 ± 6.32 µm (AZ61 + 4 wt.% Zn). Figure 5b,c illustrates the microstructures of AZ61-ECAP and AZ61 + 4 wt.% Zn-ECAP alloys, respectively. Compared to the T4 heat treatment alloy, their grains are significantly refined to average sizes of 17.38 ± 5.12 µm and 13.36 ± 3.75 µm, respectively. OM images (Figure 4) show that some parent grains in the alloy after the ECAP process are surrounded by small DRX grains, with the volume fraction of DRX grains increasing with the addition of 4 wt.% Zn, resulting in the smallest average grain size.

Figure 6 presents the SEM analysis and EDS analysis of as-cast AZ61 (Figure 6a) and AZ61 + 4 wt.% Zn (Figure 6b). The images indicate that the second phase in AZ61 without added zinc is mostly Mg-Al (highlighted by red arrows), forming clustered spherical shapes that are unevenly distributed. In contrast, AZ61 + 4 wt.% Zn exhibits a Mg-Al second phase and a Mg-Zn phase (highlighted by red arrows), presenting long and slender shapes that are uniformly distributed along the grain boundaries. These evenly distributed structures influence the strength of the metallic alloy [27].

### 3.3. Mechanical Properties

The stress–strain curves of AZ61 and AZ61 + 4 wt.% Zn under as-cast, T4, and ECAP conditions are presented in Figure 7a,b. As indicated in Table 3, the T4 and ECAP process did not significantly affect the ductility of the materials, but there were variations in both the yield strength (YS) and ultimate compression strength (UCS). The T4 homogenization treatment slightly decreased the mechanical properties of AZ61, reducing the YS (115.42 MPa) and UCS (353.97 MPa) by 7.42% and 4.17%, respectively. After ECAP treatment, the YS (141.53 MPa) and UCS (400.48 MPa) of AZ61 significantly improved compared to the as-cast alloys, increasing by 13.52% and 8.43%, respectively.

For AZ61 + 4 wt.% Zn, the T4 treatment decreased the YS (117.42 MPa) and UCS (365.43 MPa) by 15.25% and 4.94%, respectively. After the ECAP process, the mechanical properties of AZ61 + 4 wt.% Zn were maximized, with YS (145.88 MPa) and UCS (421.79 MPa) increasing by 5.29% and 9.72%, respectively. The results of compression tests indicated that the ECAP process had a greater influence on the mechanical properties of the materials than did the T4 homogenization treatment.

Figure 7c and Table 3 demonstrate a slight decrease in hardness after T4 heat treatment for both materials, with AZ61 + 4 wt.% Zn showing a decrease of 18.29%. Figure 4d,e illustrates the substantial incorporation of the β-phase into the α-phase resulting from T4 heat treatment in AZ61 + 4 wt.% Zn. After ECAP, the significant grain refinement in AZ61, as shown in Figure 7d, leads to a hardness increase to 77.44 HV.

### 3.4. Corrosion Properties

The results of salt spray tests are depicted in Figure 8. Figure 8a shows poor corrosion resistance on the surface of AZ61-as-cast, with uneven surface corrosion and numerous corroded pits (highlighted by yellow circles). Figure 8b,c displays the uneven corrosion on AZ61 after T4 heat treatment and ECAP. Figure 8d–f shows specimens of AZ61 + 4 wt.% Zn after different processing steps. With the addition of 4 wt.% Zn, the corrosion area becomes more uniform, with only a few corroded pits observed in Figure 8d AZ61 + 4 wt.% Zn-as-cast. The best corrosion resistance is observed in Figure 8f for AZ61 + 4 wt.% Zn after ECAP, with a majority of the surface remaining uncorroded and exhibiting a bright appearance.

## 4. Discussion

In this study, the influences of zinc addition and secondary processing on the mechanical properties of the AZ61 magnesium alloy have been thoroughly investigated. The initial observations revealed the presence of aluminum (Al) in the form of β-Mg_17_Al_12_ precipitates within the magnesium alloy matrix [28,29], as illustrated in Figure 4a. During the homogenization process, an increase in temperature led to an enhanced solubility of aluminum in the magnesium matrix. The β-Mg_17_Al_12_ phase was observed at grain boundaries, which eventually dissolved into the α-Mg supersaturated solid solution and cooled in the form of fine particles [30]. The second phase significantly affected the mechanical and corrosion properties of magnesium alloy [31], as confirmed by the findings presented in Figure 6b.

The addition of 4 wt.% zinc, as depicted in Figure 6b, induced structural modifications in the magnesium alloy, impacting the lattice structure and solid solubility [32]. This transformation resulted in a shift from coarse to elongated structures and a more uniform distribution of the Mg-Zn phase and second phase [33]. These phenomena significantly influenced the alloy’s mechanical performance, corrosion resistance, and other properties, as evidenced in Table 3.

Upon subjecting the alloy to T4 heat treatment, dual-grain structures were created. The heat-treatment process led to recrystallization or growth of grains, with the formation of dual-grain structures due to uneven growth rates in different directions during solidification [34]. Additionally, the presence of impurity cores within the material during heat treatment possibly guided grain orientation, contributing to the observed dual-grain structures [23]. T4 heat treatment adversely affected the mechanical properties of the AZ61 magnesium alloy, resulting in a decrease in both strength and hardness, as outlined in Table 3.

The strength behavior of materials processed through Severe Plastic Deformation (SPD), such as Equal Channel Angular Pressing (ECAP), is primarily influenced by grain size, dislocation density, and crystallographic texture [35,36,37]. The reduction in grain size, as illustrated in Figure 7d, showing post-ECAP processing of AZ61 + 4 wt.% Zn, demonstrated a pronounced grain-refinement effect, effectively enhancing mechanical strength. The improved strength can be attributed to Orowan strengthening and the load-carrying effect [38]. The heat treatment performed during the ECAP process facilitated the complete dissolution of β-Mg_17_Al_12_ into the α phase, as shown in Figure 4c,f, contributing to the enhancement of material mechanical properties.

The hardness test results in Figure 6c align with the research conducted by S.J Huang et al. [39]. The authors investigated the impact of ECAP on grain refinement in AZ61 magnesium alloys, demonstrating that an increase in the number of ECAP passes leads to a reduction in average grain size, accompanied by an increase in both hardness and ductility. Therefore, the observed hardness outcomes may be attributed to grain refinement, the size of hard phase particles, and strain hardening during the ECAP process.

This study explored the impacts of the presence, morphology, and distribution of the second phase in magnesium alloys on corrosion resistance (Figure 8). In Figure 8a,d, it is shown that the surface of AZ61-as-cast exhibits severe collapse, indicating a high corrosion rate during the salt spray test. After the addition of 4 wt.% Zn, a uniform corroded surface is formed. Dark corrosion pits (highlighted by yellow circles) are observed, indicating localized corrosion attacks during the salt spray test. It is noteworthy that the presence of pits is detrimental to overall corrosion resistance, and the degradation of AZ61 + 4 wt.% Zn-as-cast will proceed through one or more pit corrosion [40]. The AZ61 + 4 wt.% Zn alloy exhibits milder pit corrosion, while the AZ61 alloy accompanies severe collapse across the entire surface, indicating the slowest corrosion rate for AZ61 + 4 wt.% Zn alloy, whereas AZ61 alloy exhibits significant non-uniform corrosion.

Post-ECAP treatment, AZ61 + 4 wt.% Zn exhibited increased corrosion resistance, potentially associated with the size and distribution of second-phase particles [33]. The transformation of large, coarse second-phase particles to smaller sizes and a more uniform distribution after ECAP treatment was crucial for improved corrosion resistance [41]. The role of ECAP in influencing these microstructural features, especially the small and uniformly distributed second phase, proved essential in mitigating the micro-galvanic effect. Moreover, the significant reduction in grain size compared to cast alloy further enhanced corrosion resistance (Figure 4f). As grain size decreases, the density of lattice defects, such as grain boundaries and high dislocation density, increases, leading to an increase in the fraction of grain boundaries. Studies have shown that high-density lattice defects, such as grain boundaries and high dislocation density, are more likely to form a dense and uniform passivation layer [42]. These high-density lattice defects will enhance the stability and integrity of the passivation film, thereby improving the corrosion resistance of the alloy.

In conclusion, our study highlights the efficiency of alloying element addition and secondary processing in tuning the microstructure of magnesium alloys, leading to improved mechanical performance and corrosion resistance. The findings underscore the potential for tailoring magnesium alloy properties for specific applications through controlled alloying and processing techniques.

## 5. Conclusions

In this research, the stir-casting method was employed to fabricate AZ61 and AZ61 + 4 wt.% Zn magnesium alloy. The influence of Equal Channel Angular Pressing (ECAP) on the microstructure, mechanical properties, and corrosion resistance of the materials was thoroughly investigated. The key findings and conclusions are summarized as follows:(1)Microstructural analysis revealed that the addition of 4 wt.% Zn resulted in a significant refinement of grain size and promoted the uniform distribution of Mg-Zn phases and β-Mg_17_Al_12_ phases along grain boundaries. This refinement was further enhanced after ECAP processing, leading to a more pronounced grain refinement in AZ61 + 4 wt.% Zn. This refinement played a crucial role in the notable improvement of mechanical and corrosion properties.(2)The region with the most extensive Dynamic Recrystallization (DRX) grain refinement was observed in AZ61 + 4 wt.% Zn after ECAP processing, showcasing the highest yield strength (145.88 MPa), ultimate compression strength (421.79 MPa), and hardness (84.83 HV).(3)T4 heat treatment demonstrated a significant impact on the solid solution of Mg-Zn phases and β-Mg_17_Al_12_ phases in Mg-Al-Zn alloys. Especially in AZ61 + 4 wt.% Zn, the Mg-Zn phases are completely dissolved in the Mg alloy during the T4 heat treatment.(4)ECAP has been demonstrated as an effective method for enhancing the mechanical properties and corrosion resistance of magnesium alloys. The results indicate that ECAP has increased the mechanical properties of AZ61 + 4 wt.% Zn, showing improvements of 5.29% in YS and 9.72% in UCS. Furthermore, with regard to corrosion properties, ECAP significantly reduced the corrosion rate and promoted a more uniform corrosion surface.

## Figures and Tables

**Figure 1 materials-17-01678-f001:**
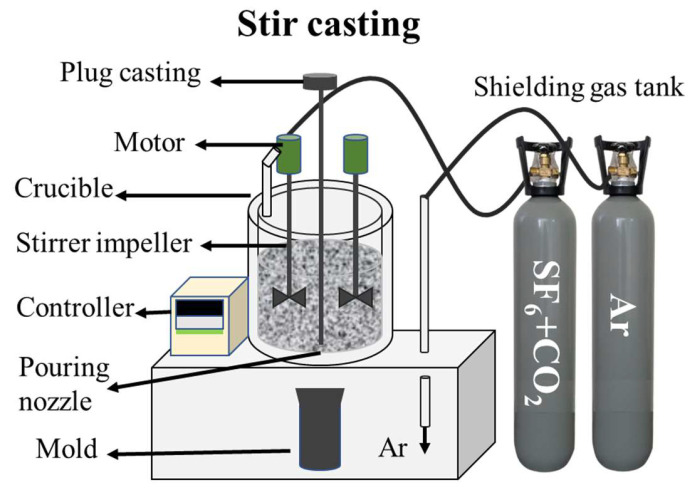
Experimental setup of the stir-casting furnace used for fabrication of the Mg ingot.

**Figure 2 materials-17-01678-f002:**
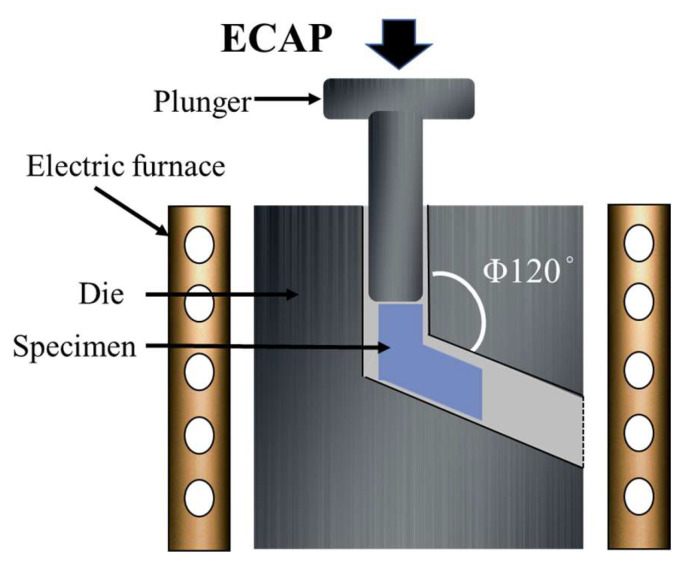
Equal Channel Angular Pressing (ECAP) setup using 120 degrees.

**Figure 3 materials-17-01678-f003:**
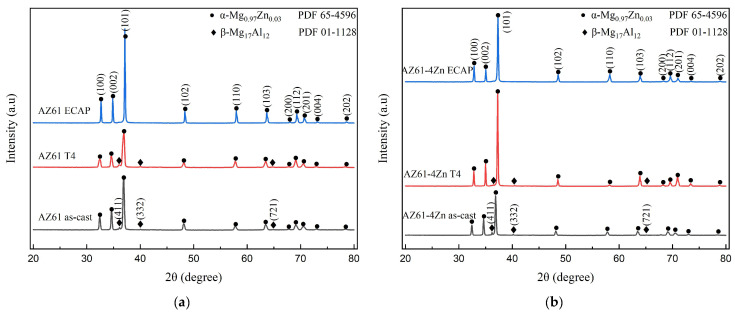
XRD analysis of fabricated (**a**) AZ61 and (**b**) AZ61 + 4 wt.% Zn.

**Figure 4 materials-17-01678-f004:**
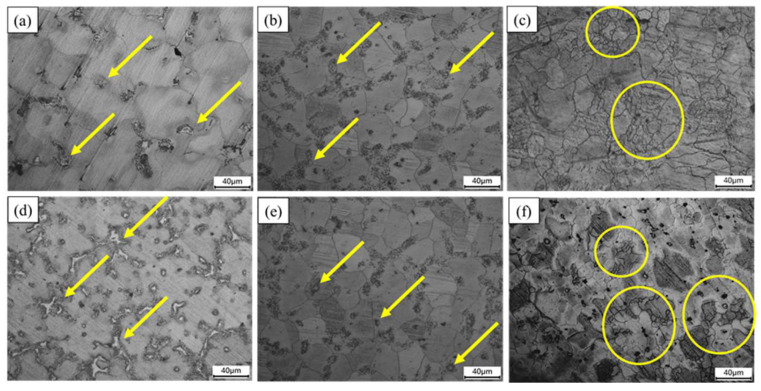
Optical microscopy microstructures of (**a**) AZ61-as-cast, (**b**) AZ61-T4, (**c**) AZ61-ECAP, (**d**) AZ61 + 4 wt.% Zn-as-cast, (**e**) AZ61 + 4 wt.% Zn-T4, and (**f**) AZ61 + 4 wt.% Zn-ECAP. The yellow arrows indicate phase particles, including β phase particles and Mg-Zn phase particles. The yellow circles indicate areas of grain refinement.

**Figure 5 materials-17-01678-f005:**
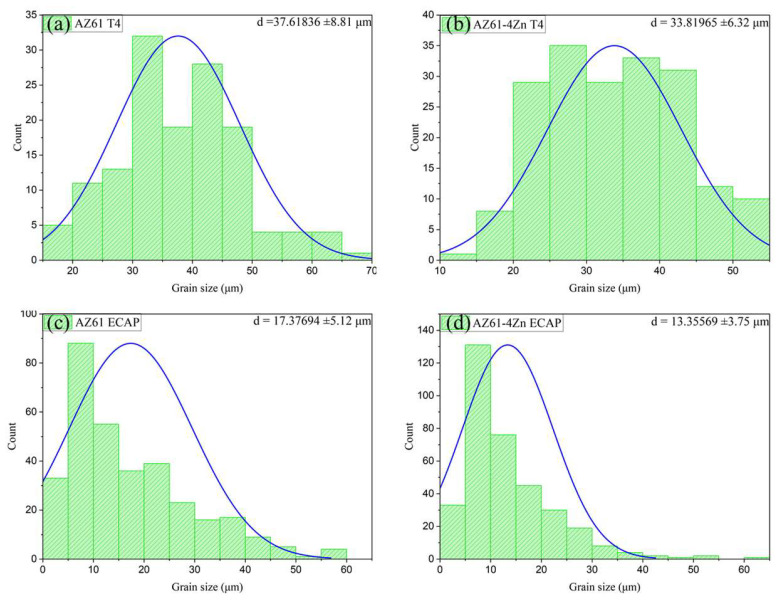
Grain size distribution of (**a**) AZ61-T4, (**b**) AZ61 + 4 wt.% Zn-T4, (**c**) AZ61-ECAP, and (**d**) AZ61 + 4 wt.% Zn-ECAP.

**Figure 6 materials-17-01678-f006:**
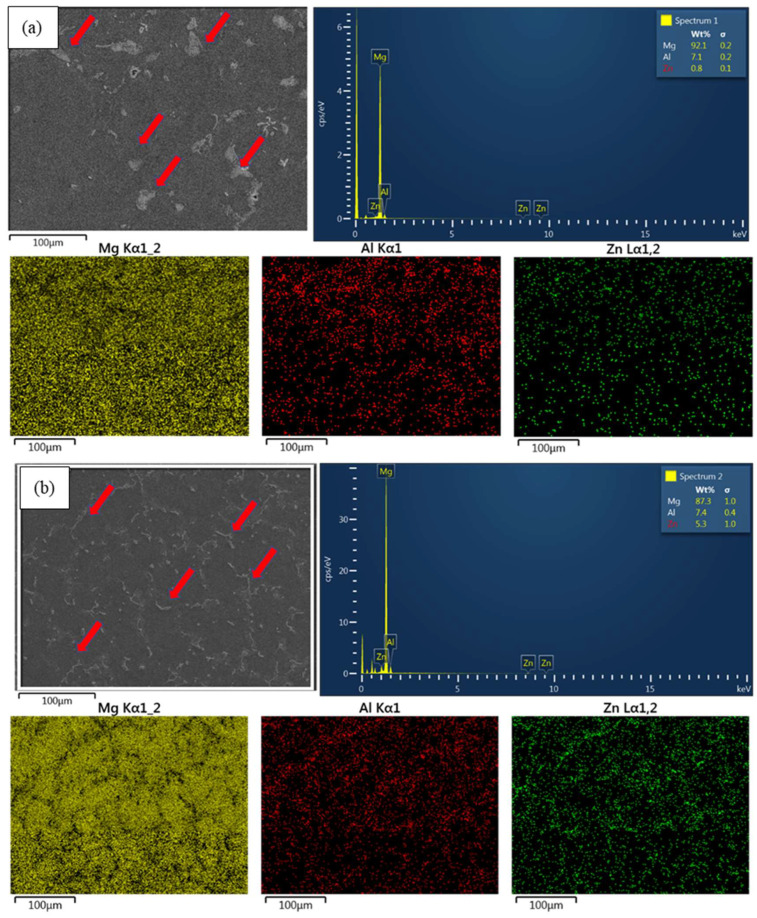
SEM microstructures and EDS elemental mapping study of (**a**) AZ61-as-cast and (**b**) AZ61 + 4 wt.% Zn-as-cast. The red arrows indicate phase particles, including β-phase particles and Mg-Zn-phase particles.

**Figure 7 materials-17-01678-f007:**
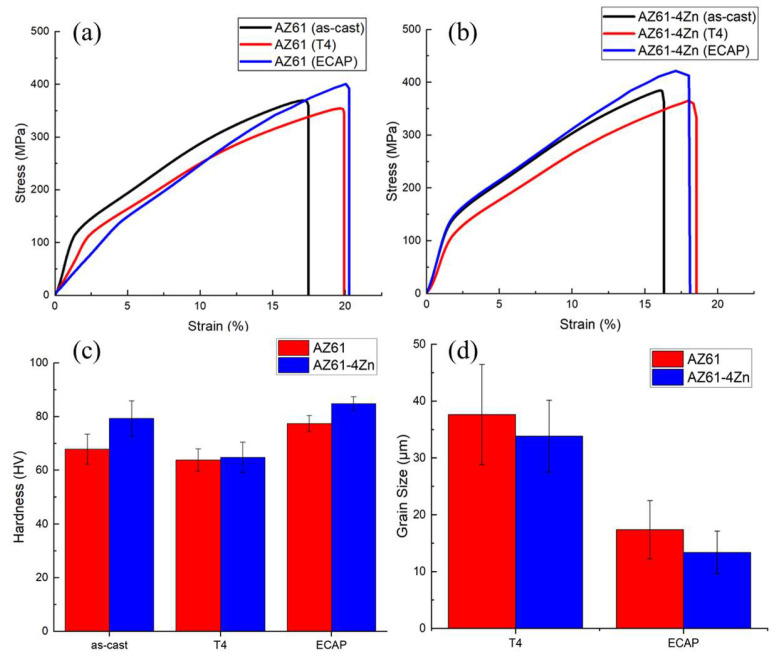
Stress–strain curves of (**a**) AZ61 and (**b**) AZ61 + 4 wt.% Zn; (**c**) Vickers hardness; and (**d**) Average grain size.

**Figure 8 materials-17-01678-f008:**
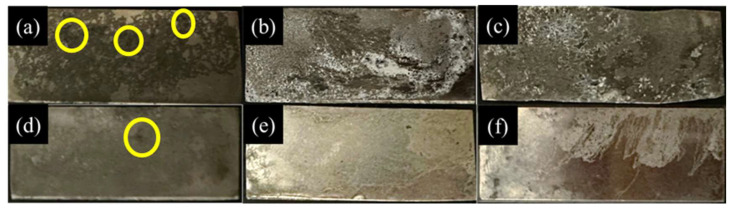
Surface morphology of salt-spray-treated samples of (**a**) AZ61-as-cast, (**b**) AZ61-T4, (**c**) AZ61-ECAP, (**d**) AZ61 + 4 wt.% Zn-as-cast, (**e**) AZ61 + 4 wt.% Zn-T4, and (**f**) AZ61 + 4 wt.% Zn-ECAP. The yellow circles indicate the areas of localized corrosion.

**Table 1 materials-17-01678-t001:** The chemical composition of AZ61.

Elements	Mg	Al	Zn	Mn	Si	Fe	Cu	Ni
wt.%	balance	6.25	1.24	0.27	0.06	0.03	0.01	0.01

**Table 2 materials-17-01678-t002:** Etching solution combination.

Ethanol (mL)	DI Water (mL)	Acetic Acid (mL)	Picric Acid (g)	Time (s)
100	10	5	6	25

**Table 3 materials-17-01678-t003:** A summary of the mechanical characteristics of the AZ61 and AZ61 + 4 wt.% Zn alloy.

Metallic Alloys	Process	EL (%)	YS (MPa)	UCS (MPa)	Hardness (HV)
AZ61	as-cast	19.26 ± 4.24	124.67 ± 3.60	369.36 ± 4.24	67.88 ± 5.61
	T4	21.83 ± 2.60	115.42 ± 5.05	353.97 ± 7.71	63.87 ± 4.14
	ECAP	22.96 ± 2.84	141.53 ± 3.80	400.48 ± 4.10	77.44 ± 3.12
AZ61 + 4 wt.% Zn	as-cast	18.49 ± 4.88	138.55 ± 4.14	384.43 ± 9.55	79.33 ± 6.63
	T4	19.73 ± 2.50	117.42 ± 3.40	365.43 ± 6.74	64.82 ± 5.74
	ECAP	20.58 ± 3.66	145.88 ± 3.14	421.79 ± 4.25	84.83 ± 2.62

## Data Availability

The data presented in this study are available in the article.

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
