# Peer review of "Effect of Zinc and Severe Plastic Deformation on Mechanical Properties of AZ61 Magnesium Alloy"

_materials, 2024, doi:10.3390/ma17071678_

Round 1

Reviewer 1 Report

Comments and Suggestions for Authors

Notes on the article of Song-Jeng Huang, Sheng-Yu Wu and Murugan Subramani «Effect of Zinc and Severe Plastic Deformation on Mechanical Properties of AZ61 Magnesium Alloy»

The paper reports about the effect of Zn addition and equal channel angular pressing (ECAP) on the structure, mechanical properties and corrosion resistance of magnesium AZ61 alloy. The authors showed that Zn affect the structure, mechanical and corrosion properties of the AZ61 alloy. Unfortunately, the article is of rather low quality, and some of the presented data and discussion are questionable. In particular, it is not clear what justifies the choice of the number of ECAP passes and the method of manufacturing the alloy. In its present form, the article does not meet the requirements for scientific articles, but is more like a technical report.

1.                 What is the point of adding zinc particles to the alloy? Zinc has a lower melting point than compared to melting point of magnesium AZ61 alloy. Alloys are usually reinforced with more refractory particles. Why didn't the authors melt a magnesium alloy with a high zinc content?

2.                 The authors should describe in more detail the experiments to determine the mechanical properties and corrosion resistance. At a minimum, they should indicate how many samples were used to determine the value of the parameter.

3.                 It is not clear why the authors made only one ECAP pass. It is known that the structure formed after one ECAP pass consists predominantly of grains with low-angle boundaries. A more stable structure is formed after several ECAP passes (for example, for route Вс after 4) [Langdon, T.G. The principles of grain refinement in equal-channel angular pressing // Materials Science and Engineering A. - 2007. – V. 462. – P. 3 – 11]. In this case, the structure is also heterogeneous after ECAP, and the increase in mechanical properties is small, which indicates the advisability of increasing the number of passes.

4.                 The authors state: «Moreover, after undergoing ECAP with one pass, only the α - Mg0.97Zn0.03 phase was observed, indicating homogenization and complete dissolution of the β-Mg17Al12 phase during the ECAP process». The dissolution of particles occurs not only during deformation. Before ECAP, the sample was heated at a temperature close to the homogenization temperature. Some particles could have dissolved during heating.

5.                 It is doubtful that with such a Zn content in the cast alloy AZ61+4%Zn, zinc particles or phase do not precipitate (Figure 3).

6.                 When the authors discuss the effect of deformation on properties, they talk about structure and do not mention texture. Although, judging by Figure 3, the texture of the alloys change. For example, the line (101) becomes more intense in the AZ61 alloy after ECAP. At the same time, the (101) line weakens (compared to the state after T4), and the (100) line becomes more intense than the (002) line in the AZ61+4%Zn alloy after ECAP.

7.                 The authors state: «As shown in Figure 6d after ECAP, the grain size reduces 70.17% in AZ61+4wt.%Zn». The authors' conclusion does not agree well with Figure 4 c and f. In these pictures we can see many grains with a size of about 40 microns. The authors should more carefully calculate the grain size and provide a grain size distribution.

8.                 Figure 4. An explanation of what the arrows in the figure mean should be added to the caption. In the case of Figures 4 b and e, it is not clear what the arrows indicate.

9.                 The structure presented in Figure 4 is extremely poorly prepared. Doubts arise that all dark zones along the boundaries are a phase. This is more like an etching defect (over-etching). In addition, the authors claim that an increase in zinc content led to an increase in the volume fraction of the phase. However, according to the XRD results, no zinc phase is formed in the AZ61+4wt.%Zn alloy. Then how could an increase in zinc increase the volume fraction of the phase? Also, the phase has a different configuration  in Figure 5b.

10.            The authors state: «In contrast, AZ61+4wt.%Zn exhibits a Mg-Al-Zn second phase (highlighted by red arrows), presenting long and slender shapes uniformly distributed along the grain boundaries». Previously, the authors stated that the zinc phase is not formed.

11.            «Stress-strain curves for AZ61 and AZ61+4wt.%Zn under…… improvements of 6.85%, 17.02%, and 14.19%, respectively». Discussion about changes in mechanical characteristics is questionable. The authors did not indicate the statistical error in determining these characteristics in order to discuss their increase.

12.            Figure 6 a. Why does the slope of stress–strain curves change?

13.            Figure 6 d. See note 6.

14.            Table 2. Statistical error value should be added.

15.            Table 2 presents the results of studies of the mechanical characteristics of alloys. Earlier in the abstract the authors wrote: «The attained mechanical properties demonstrate that after a single ECAP pass, AZ61 + 4wt.%Zn alloy exhibits optimal yield strength (YS), ultimate compression strength (UCS), and hardness». It can hardly be called optimal with such a ratio YCS and UCS values. The authors should clarify what they mean by the word «optimal».

16.            The results of the corrosion resistance study are also questionable. Firstly, it is not clear based on what considerations the authors chose the zone on the corroded samples for comparison. Why is only one side of one sample shown? Why don't the authors calculate the value of corrosion rate?

17.            Figure 7. An explanation of what the circles in the figure mean should be added to the caption.

18.            The authors discuss: «Post-ECAP treatment, AZ61+4wt.%Zn exhibited increased corrosion resistance, potentially associated with the size and distribution of second-phase particles [30]». However, in the case of the alloys considered in the article, these arguments are questionable. Firstly, according to the authors, no particles are formed in alloys after ECAP. Secondly, if this is the case, then it is not clear why the corrosion resistance of the cast AZ61+4wt.%Zn alloy is higher than that of the AZ61 alloy, despite the increase in the phase fraction.

19.            «Moreover, the significant reduction in grain size compared to cast alloy further enhanced corrosion resistance (Figure 4f)». Typically, the grain refinement reduces corrosion resistance due to an increase in the extent of boundaries and the proportion of dislocations.

20.             Line 79: it should be written « add SF6 + CO2 mixture » instead of « add SF6 + CO2 mixture».  Line 101: «channel angle of 120 degrees and a pressing speed of 20 mm/min..» Extra dot.

Reviewer 2 Report

Comments and Suggestions for Authors

The article "Effect of Zinc and Severe Plastic Deformation on Mechanical Properties of AZ61 Magnesium Alloy" investigates an alloy with promising tecnical applications, and in particular reports heat treatment effects and Zn content on mechanical/deformation properties.

This work is quite interesting and well-written, however
in my opinion, some minor issues regarding the experimental part should be improved before that article can be accepted for publication. I see there the following points.

1) Lines 78-84, the description of the alloy manufacturing. The whole paragraph starts with past tense, and equally
the last phrase teminates with past tense. However, all the phrases in the middle sound like commands in the present tense.
Here it is necessary to follow a unique descriptive narrative style, using past tense.

2) Lines 106 ff, the description of the X-ray diffractometer. The indicated electrical power of 300 Watts is irrelevant.
Instead, authors should indicate they have used monochromatic CuK_alpha radiation (if so), the anode current (which determines the intensity)
and the operating Voltage (already given).
Then the continuation with scan rate, etc. is o.k.

3) Text inside Figure 3 looks a little blurred and surrounded with black tiny dots. This seems to be a problem of the jpg format, so the
authors could produce these figures in pdf directly to avoid this.

Reviewer 3 Report

Comments and Suggestions for Authors

The manuscript titled “Effect of Zinc and Severe Plastic Deformation on Mechanical Properties of AZ61 Magnesium Alloy” is presenting investigations on the impact of alloying (increasing Zn content), solution heat treatment (T4 heat treatment), and severe plastic deformation (ECAP) on the strength enhancement of AZ61 magnesium alloy. This paper is not recommended for publication in the current state.

Following are only some of the issues in the manuscript which are to be seriously addressed:

1.           The English language and grammar used in the present manuscript is in patches poor. There are plenty of instances where mistakes, misspelled words and/or poorly chosen words (ambiguous) are present. I strongly suggest that the paper should be proofread and double-checked concerning the spelling and phrasing.

2.           I highly recommend including a ternary diagram (Mg-Al-Zn) in the article to visually assess the conformity of the results. The ternary diagram will provide a precise representation, allowing for a clear evaluation of the data.

3.           I have some uncertainty regarding whether the obtained material is a composite or a metallic alloy. The material's composite nature is evident despite the observed zinc composition of about 5% in the second alloy/composite material or 5.3% according to EDS area analysis, which is below the expected threshold of maximum of 6.2% for zinc dissolution in the solution. This discrepancy underscores the composite's unique composition and suggests the presence of other reinforcing elements beyond the zinc content. The matrix of the composites must be made of a material capable of incorporating the dispersed component, which cannot be destroyed by dissolution, melting, chemical reaction or mechanical action.

4.           I am interested in understanding why the β phase does not dissolve after the T4 treatment, despite the equilibrium diagram indicating that magnesium can dissolve up to about 9% aluminum at 380 °C. I noticed you mentioned that this phenomenon occurs after ECAP deformation. Could you elaborate on why this discrepancy exists?"

5.           How was the grain size measured, and why weren't measurements provided for the as-cast samples as well?

6.           What method was employed to determine the elongation? Was Hooke's law applied in the calculation process? It seems that the value wasn't directly obtained during compression, but rather derived.

7.           Why did the hardness decrease after the T4 treatment? Would not it have been expected for the hardness to increase after being put in the solution? Furthermore, could you please describe how the quenching process was conducted?

8.           It looks that the microstructure observed in the figure 4d has undergone more noticeable changes as a result of the etching process compared to other samples. The etching treatment seems to have had a more pronounced effect on the microstructural features, possibly revealing different phases or structural details that are more prominent in this figure. Are you sure that the attack was carried out properly?

9.           I kindly request to recalculate the increase/improvement rates for the properties outlined in the conclusions. The current calculations appear to be inaccurate.

• Overall, I consider that the paper can be considered for publication after major correction.

Comments on the Quality of English Language

Round 2

Reviewer 1 Report

Comments and Suggestions for Authors

See additional questions in the attachment.

Author Response

Thank you so much for your valuable comment.

Reviewer 3 Report

Comments and Suggestions for Authors The authors have adequately addressed most of my concerns, in the revised version of the manuscript. Therefore, I have no further comments, except for the following:   I think there was a confusion regarding the type of material obtained. I tried to highlight the reasons why it cannot be considered a composite. I did not say that it is a composite material. Even if I agree with your answer, namely, this material is a metallic alloy, the fact that you obtained a composite is still written in several places in the last version of manuscript. It seems that your answer contradicts the content of the paper.    

Author Response

(The authors gave the same response as above.)
